# Modeling Community-Scale Natural Resource Use in a Transboundary Southern African Landscape: Integrating Remote Sensing and Participatory Mapping

Kyle D. Woodward [1,*], Narcisa G. Pricope [1], Forrest R. Stevens [2], Andrea E. Gaughan [2], Nicholas E. Kolarik [2], Michael D. Drake [3], Jonathan Salerno [4], Lin Cassidy [5], Joel Hartter [3], Karen M. Bailey [3] and Henry Maseka Luwaya [6]

1.  Department of Earth and Ocean Sciences, University of North Carolina Wilmington, 601 S College Road, Wilmington, NC 28403, USA; pricopen@uncw.edu
2.  Department of Geography and Geosciences, Lutz Hall, University of Louisville, Louisville, KY 40292, USA; forrest.stevens@louisville.edu (F.R.S.); ae.gaughan@louisville.edu (A.E.G.); nicholaskolarik@u.boisestate.edu (N.E.K.)
3.  Environmental Studies Program, Sustainability, Energy, and Environment Community, University of Colorado Boulder, 4001 Discovery Drive, Boulder, CO 80303, USA; Michael.Drake-1@colorado.edu (M.D.D.); joel.hartter@colorado.edu (J.H.); Karen.bailey@colorado.edu (K.M.B.)
4.  Department of Human Dimensions of Natural Resources, Graduate Degree Program in Ecology, Colorado State University, Campus Box 1480, Fort Collins, CO 80523-1480, USA; jonathan.salerno@colostate.edu
5.  Okavango Research Institute, University of Botswana, P/Bag 285, Maun, Botswana; lcassidy@ub.ac.bw
6.  Department of National Parks and Wildlife, Private Bag 1, Kafue Road, Chilanga, Zambia; henrymaseka@gmail.com
*   Correspondence: kdwoody11@gmail.com

**Abstract:** Remote sensing analyses focused on non-timber forest product (NTFP) collection and grazing are current research priorities of land systems science. However, mapping these particular land use patterns in rural heterogeneous landscapes is challenging because their potential signatures on the landscape cannot be positively identified without fine-scale land use data for validation. Using field-mapped resource areas and household survey data from participatory mapping research, we combined various Landsat-derived indices with ancillary data associated with human habitation to model the intensity of grazing and NTFP collection activities at 100-m spatial resolution. The study area is situated centrally within a transboundary southern African landscape that encompasses community-based organization (CBO) areas across three countries. We conducted four iterations of pixel-based random forest models, modifying the variable set to determine which of the covariates are most informative, using the best fit predictions to summarize and compare resource use intensity by resource type and across communities. Pixels within georeferenced, field-mapped resource areas were used as training data. All models had overall accuracies above 60% but those using proxies for human habitation were more robust, with overall accuracies above 90%. The contribution of Landsat data as utilized in our modeling framework was negligible, and further research must be conducted to extract greater value from Landsat or other optical remote sensing platforms to map these land use patterns at moderate resolution. We conclude that similar population proxy covariates should be included in future studies attempting to characterize communal resource use when traditional spectral signatures do not adequately capture resource use intensity alone. This study provides insights into modeling resource use activity when leveraging both remotely sensed data and proxies for human habitation in heterogeneous, spectrally mixed rural land areas.

**Keywords:** remote sensing; participatory mapping; NTFP; grazing; random forest; natural resources; drylands; savanna woodlands

## 1. Introduction

Rural communities in southern Africa face a variety of climatic and environmental challenges that contribute toward their vulnerability. People in this region often rely on rain-fed agriculture for their livelihoods. However, fluctuations in inter- and intra-annual rainfall patterns are increasing in their intensity and duration [1–4], contributing to crop yield sensitivity [5–7]. Crop loss from wildlife, pests, and disease can further constrain food and economic resources for a household [8–10]. To overcome such hardships, diversify their livelihood base, and buffer themselves from climatic shocks, many households raise livestock and collect natural resources from surrounding lands [11–15]. However, the natural resource bases that allow for these additional livelihood activities are also susceptible to overuse through natural resource exploitation and environmental changes.

Local natural resources, frequently termed non-timber forest products (NTFPs), significantly contribute to daily livelihood needs and income of Africans, both across the rural–urban divide [16–20] and across income levels [11,21,22]. NTFPs are infinitely diverse in their sources [23], however, common uses include food, medicine, cooking fuel, and materials for household construction and marketable craftwork [24–26]. Along with productive grazing areas for livestock, NTFP collection areas are significant components of the land systems and are critical for supporting rural livelihoods in southern Africa. However, land conversion to cropland and settlements as well as overexploitation of naturally occurring resources are a clear hindrance to the sustainability of the natural resource base in southern Africa [27–29]. Increasing climate variability may further exacerbate these issues by constraining seasonal and long-term accessibility of NTFPs and productive grasslands [30–33].

Given the socioeconomic importance of NTFPs and livestock ownership in rural African communities, it is imperative to know where on the landscape people engage in these activities and to what level of intensity people use valuable areas. Analyzing the spatial and quantitative components to resource use is critical to uncovering the social, economic, and environmental patterns that influence natural resource use decisions at different levels of social organization. To effectively make those linkages, a people and pixel approach provides a path forward that leverages remote sensing data with ground-based social data for capturing specific resource use activities on the ground [34,35]. Previous work shows that remote sensing integrated with socio-ecological research can help to elucidate drivers of resource use and land cover change [36–40], identify environmental causes of socioeconomic challenges [41], validate local environmental perceptions [42,43], and produce relevant maps of locally important ecotypes and places [44–46].

One approach to mapping natural resource use activity is through participatory rural appraisals (PRA), a suite of participatory methods such as household and individual surveys, focus groups, workshops, and participatory mapping [47]. The combination of participatory mapping and geographic information systems (GISs) has been integral in mapping and analyzing spatial characteristics of the human–environment landscape [48], and increasingly, the use of remote sensing has become more tractable (i.e., finer spatial and temporal grains) and accessible in such participatory efforts [34,49–51].

While remote sensing has been used in PRA contexts to answer questions related to grazing [52–54], few studies relate to communally held land tenure systems such as community-based organizations (CBOs) (but see [55,56]). Remote sensing has also been integrated into PRA studies that aim to map NTFP availability, to analyze ecosystem effects of NTFP collection, and to understand the collection and use behaviors of the community members who rely on them [57,58]. However, NTFP studies that incorporate remote sensing techniques beyond manual interpretation of satellite imagery more commonly focus on individual plant species with NTFP value [59–61]. Of those remote sensing studies that do focus on broader NTFP functional scale (firewood, building materials, etc.), only a few use remote sensing techniques such as vegetation indices and supervised classification to model or characterize human use of NTFPs [62,63]. Therefore, there is

much to be investigated regarding the capabilities of remote sensing analyses in modeling NTFP collection and grazing activities in communally managed landscapes.

In this study, we used a GIS-based resource area dataset that was previously produced using participatory mapping methods [64] to map the intensity of three distinct types of resource use: livestock grazing, firewood collection, and building pole collection. The main objective of this study was to investigate the feasibility of integrating participatory mapped training data with multisource remotely sensed data within a commonly used machine learning classification model. The random forest (RF) approach is a nonparametric, ensemble classification technique that has become quite common in land cover applications [65,66], with its robustness preferable or on par with other classification algorithms [67]. RF classifiers lend themselves well to high-dimensional data [65] and their insensitivity to highly correlated and non-informative variables makes them appealing to use with various bands of spectral data and texture features [66,68]. The ability to combine discrete and continuous data as model inputs makes RF attractive for interdisciplinary modeling approaches that draw from a variety of data sources [69]. RF implementations are also desirable due to their few critical tuning parameters and they require less time and computational resources to train compared to other machine learning models [70]. The random forest internal variable importance assessment, made available through "bagging" of training data [71], not only aids in improving model performance, but also helps practitioners understand which variables are most informative for classifying the phenomena of interest [72].

We apply this methodology in the Kavango–Zambezi Transfrontier Conservation (KAZA) region of southern Africa, where many rural communities are predominantly natural resource-dependent and local political institutions commonly referred to as community-based organizations (CBO) have been established to facilitate community-based natural resource management and sustainable development [73–75]. Input data for the model include various Landsat-derived vegetation indices and texture features as well as ancillary variables that represent human population patterns and mobility. Given their temporal and spatial coverage, as well as their accessibility, Landsat data remain a good option for regional remote sensing assessments of land conditions and offer the capability of long time series due to the continuous operation of satellite missions [76]. The open-access population proxy variables (100 × 100 m at equator) are derived from the WorldPop Project and are processed under standardized protocols [77]. We address three research questions: **(RQ1)** Can Landsat satellite remote sensing data reliably contribute toward accurate prediction of resource use intensity when trained with data derived from participatory mapping? **(RQ2)** How useful are population proxy variables for predicting resource use intensity in this region? **(RQ3)** How do prediction performance and predicted resource use intensity patterns compare between resource types and communities? Although remotely sensed data have been used to map and quantify valuable natural resources in communally managed areas [46,50,78], we are unaware of any attempt to spatially characterize the intensity at which communities engage in these subsistence resource use activities. Such information is critical in understanding how environmental and human factors may influence subsistence-based livelihood activities now and in the future.

## 2. Study Area

We focus our analysis on a transboundary region of southern Africa encompassing three CBOs. The Chobe Enclave Conservation Trust in Botswana (CECT, population: 4108; area: 2990 km$^2$) was established in 1994 [79,80]. The Mashi Conservancy in Namibia (LWZ GMA, population: 2000; area: 297 km$^2$) became an official CBO in 2003. The Lower West Zambezi Game Management Area (population: 70,157; area: 19,300 km$^2$) was established in 1971, however, it was not established as a CBO until the 1990s [81]. While each CBO is adjacent to a major river and has river-adjacent and inland villages, Mashi Conservancy and the LWZ GMA are similar in that the more remote villages lie further inland from the river, while the opposite occurs in CECT (Figure 1)**.** People in these communities raise

livestock and rely on collected natural resources for daily household needs such as fuel, construction materials, wild foods, and medicines.

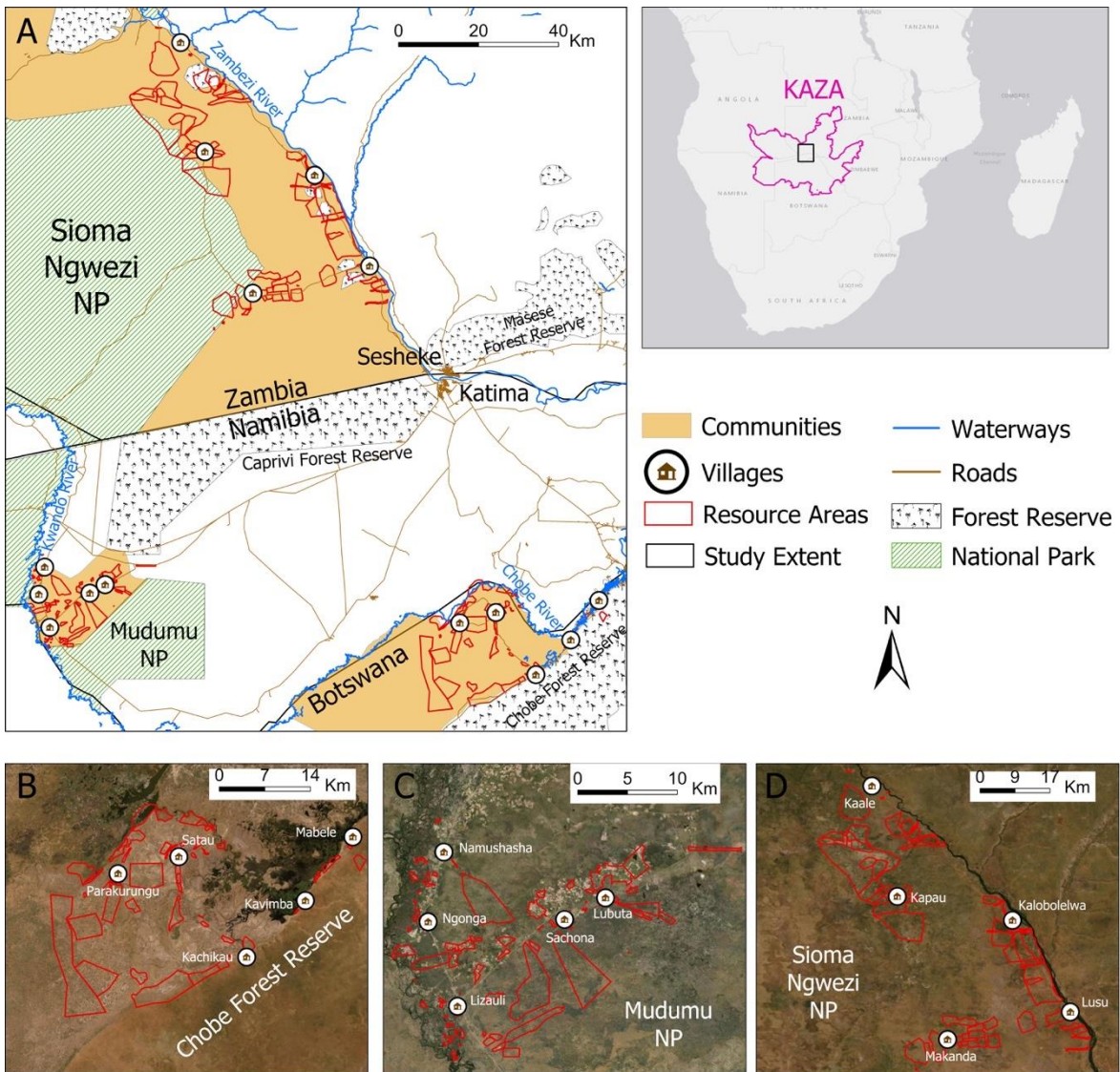

**Figure 1.** Illustration of the full modeling extent, with mapped resource areas (**A**), as well as the locations of surveyed villages within the Chobe Enclave Conservation Trust (**B**), Mashi Conservancy (**C**), and the Lower West Zambezi Game Management Area (**D**).

According to the European Space Agency's 2018 Land Cover Product [82] (Figure S1), shrub and herbaceous vegetation cover are most abundant in CECT, while Mashi and the Lower West Zambezi GMA have larger areas of forest cover. The Lower West Zambezi GMA comprises the highest density of both closed and open forest cover, while Mashi contains the largest proportion of classified cropland cover (Figure S1; Table A1 (Appendix A)).

## 3. Data

### 3.1. Field Data

We conducted household surveys and participatory mapping exercises in each CBO and randomly sampled 5 villages in each (*n* = 240 surveys × 3 CBO areas) for semi-structured surveys during the 2016, 2017, and 2018 dry seasons. Among other questions

related to food security and adaptive capacity, we asked respondents the following about NTFP collection and livestock grazing activities:

1.  "Name of the main area your livestock usually grazes in the wet/dry season in the past 3 years?"
2.  "Did you collect any of the following natural resources during the wet/dry season in the past 3 years?" (Listed resources were firewood, thatching grass, building poles, fish, reeds, palm leaves, medicinal plants, and fruits and vegetables.)
3.  "[What is the] Name of the main area where you usually collected [the resource] in the wet/dry season in the past 3 years?"
4.  "How do you usually travel there (walk, cart, canoe, etc.)?"
5.  "Time taken to get to the area?"
6.  "Total quantity gathered during the wet/dry season since this time last year?"

Resource area placenames were then aggregated from completed surveys, and the resource areas whose names were frequently referenced in household surveys were prioritized for participatory mapping.

Key informants who had extensive knowledge of the area surrounding their village aided in resource area mapping. These individuals guided researchers to each resource area and gave detailed descriptions of the boundaries of each area. We recorded boundary points using a Garmin GeoTrex handheld Global Positioning System (GPS). These GPS points were then imported into ArcMap 10.6 and polygons were digitized for each resource area through using the GPS boundary points, their boundary descriptions recorded in the field, and high-resolution satellite imagery for reference. Natural resource collection metrics were computed to reflect overall resource collection activity in each CBO (Table 1). Further detail on household survey and participatory mapping research methods are provided in [64].

**Table 1.** Summary statistics for all resource areas and households by country. Average values (standard deviation) are displayed where applicable.

| CBO (Country) | Resource Areas | Average Perimeter (km) | Average Area (km²) | # Households Using Each Resource Area | # Resource Areas Used by Each Household | Average Amount Collected/Resource Area (kg) | Average Amount Collected/Household (kg) |
|---|---|---|---|---|---|---|---|
| LWZ GMA (Zambia) | 77 | 11.2 (9.7) | 10.2 (23.2) | 3.2 (7.4) | 2.2 (1.0) | 9033 (25,162) | 2117 (9658) |
| CECT (Botswana) | 53 | 7.3 (8.1) | 4.4 (10.1) | 5.8 (8.5) | 2.6 (1.4) | 907 (1351) | 180 (370) |
| Mashi (Namibia) | 84 | 3.6 (4.5) | 1.2 (3.0) | 3.3 (5.2) | 2.5 (1.2) | 2145 (5084) | 354 (1080) |
| Total | 214 | 7.3 (8.3) | 5.22 (15.3) | 3.8 (7) | 2.4 (1.2) | 4095 (15,213) | 907 (5772) |

For this study, we used the household interview and participatory mapping data to quantify household resource use in a standardized way in order to create resource use intensity labels for classification. We summed the total number of wet season and dry season use reports from all surveyed households to each resource area, referred to hereafter as the Household Use value. Since we treated wet season and dry season resource use reports separately, each contributed 1 count in the total Household Use value. This means that if 1 household engaged in a resource use activity in both the wet and dry season, a count of 2 was added to the total Household Use value for that given resource area and resource use type. If a resource area was not reported being used for a given resource by any surveyed households in either season, its Household Use value was "NA" for that resource type. Counting the wet and dry season Household Use separately ensures that the derived resource use intensity metric accounts for the seasonal variability in resource use patterns that may arise between resource use types and households. We used this Household Use

value to create resource use intensity classes for the training data in our analysis. While we did compute the Household Use value for every resource type, in this paper, we focused on modeling wood collection, building pole collection, and grazing intensity.

We also collected reference samples in the field using a Trimble Geo7x unit to validate resource use activity predictions with on-the-ground observations. A total of 198 reference sample points were recorded, 57 within Mashi, 88 within the Chobe Enclave, and 53 within the Lower West Zambezi GMA. Information recorded at each location included observed evidence of human use by type, as well as dominant land cover, canopy closure, and distance from nearest road.

### 3.2. Landsat Data

We utilized Landsat Collection 1-Level 2 surface reflectance products from USGS EarthExplorer at the onset of the dry season (May/June) for 3 distinct years (Figure S2). In determining the 3 years of satellite data to use for our analysis, we focused on capturing a time period that meaningfully coincided with the change in community-based management that occurred in our study communities from the 1990s to the present, while ensuring minimal climatological differences between image years. We chose 2018 as the most recent time step because it was the last year that we conducted fieldwork. We consulted the National Oceanic and Atmospheric Association's (NOAA) Oceanic Niño Index (ONI) [83] to evaluate which other 2 image years between 1990 and 2000, and 2000 and 2010 would be most comparable climatologically to 2018. We chose 1994 and 2004 as the other 2 image years, because they represent a time period before the establishment of communal resource management in our study communities and a time period that is mid-way to the contemporary 2018 image year. To further validate climatological comparability between years, we produced a time series of each hydrologic year using Famine Early Warning Systems Network (FEWS NET) Climate Hazards Group Infrared Precipitation with Stations (CHIRPS) dekadal rainfall totals (Figure S3) [84].

### 3.3. Population Proxy Data

The WorldPop project provides open access to spatiotemporally harmonized gridded geospatial data layers that aid in human population mapping at fine spatial scale [77,85]. The source data contributing to these various geospatial data layers are described fully in [85], but pertinent sources include Viewfinder Panoramas topography data derived from NASA Shuttle Radar Topography Mission (SRTM), the Globcover land cover product from the European Space Agency (ESA) and Universite Catholique De Louvain (UCL), MODIS MOD44W inland water bodies from University of Maryland and NASA, Landsat-derived inland water bodies from the University of Maryland, and Open Street Map (OSM) for general landscape mapping, among others.

The appeal of using these covariates is that they are standardized across countries and have been vetted for their utility in population distribution estimates [86,87]. The WorldPop global covariates are produced at 3 arc-second (100 m at equator) spatial resolution, which is a medium resolution compatible with Landsat data and is appropriate for modeling resource use activity across the transnational landscape of our study region.

Researchers commonly use geospatial covariates related to topography, land cover features, and human infrastructure in remote sensing studies that aim to map human population [88,89], land cover [90], and resource use [91]. Therefore, we used six WorldPop geospatial covariate datasets related to road infrastructure, water bodies, and topography (Table 2; Figure S4). We chose these because they collectively comprise a quantitative representation of human settlement as well as people's proximity and accessibility to their surrounding landscape.

**Table 2.** WorldPop geospatial covariates used in the model.

| WorldPop Dataset | Model Variable Name | Data Sources |
| --- | --- | --- |
| Distance to OSM Major Roads 2016 | osm_dst_road_100 m | Lloyd et al., 2017 |
| Distance to OSM Major Road Intersections 2016 | osm_dst_roadintersec_100 m | Lloyd et al., 2017 |
| Distance to OSM Major Waterways 2016 | osm_dst_waterway_100 m | Lloyd et al., 2017 |
| Distance to ESA-CCI-LC inland water per country | esaccilc_dst_water_100 m | Lamarche, C. et al., 2017 |
| SRTM-based slope per country 2000 | srtm_slope_100 m | de Ferranti, J., 2017 |
| SRTM-based elevation per country 2000 | srtm_topo_100 m | de Ferranti, J., 2017 |

## 4. Methods

### 4.1. Preprocessing

For the purposes of random forest modeling, we preprocessed our Landsat variables, population proxy variables, and resource use response variables into a stack of raster layers with a spatial resolution of 100 m/pixel. We buffered the resource area vector dataset by 5 km and used its minimum bounding box as our modeling extent, as shown in Figure 1A. Preprocessing steps were conducted with standalone R scripts, ArcGIS 10.7, and ERDAS Imagine software (Figure 2).

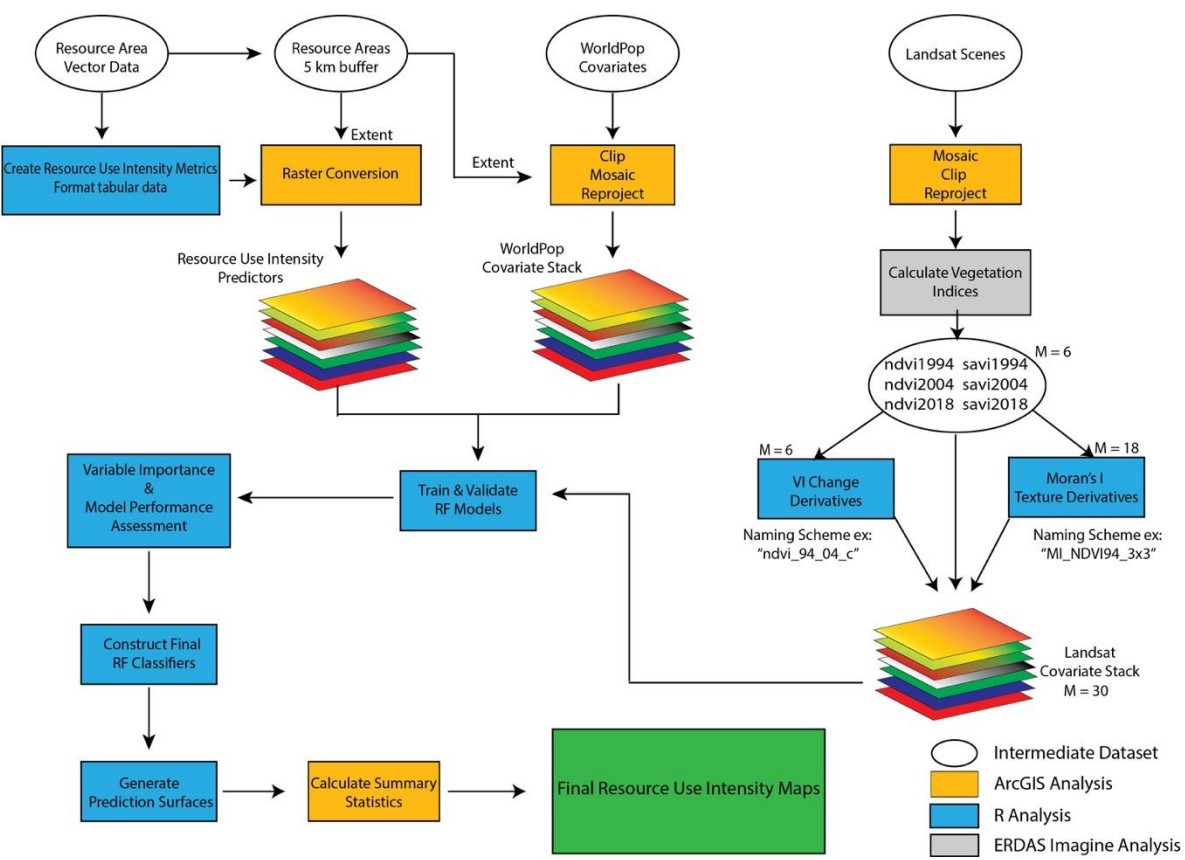

**Figure 2.** Workflow overview. Landsat-derived model variable names and variable naming schemes are given for reference.

Using the social survey-derived Household Use value described in Section 3.1, we experimented with several data standardization methods to create resource use intensity response variables for each resource type. After constructing and assessing performance of RF models in 3 formats, we chose a three-class, ordinal response variable for classification over binary and continuous formats. We chose this categorization to standardize across

resource use types while still accounting for meaningful differences in usage intensity after aggregating among households.

To create the three-class resource use intensity classes to assign to each resource area, we calculated the mean of Household Use values for each resource type. Resource areas with Household Use values above the mean were assigned to class 2, below the mean assigned to class 1, and NA values assigned to class 0. The classes are labeled as "Little to No Use" (0), "Low Use" (1), and "High Use" (2). We then assigned each resource area with its corresponding intensity label, creating a coded raster layer for each resource type (Figures 2 and 3). Class-specific summary statistics were performed on the newly aggregated data for each resource use type to illustrate distribution among class labels for resource areas in each CBO, as provided in Table A2.

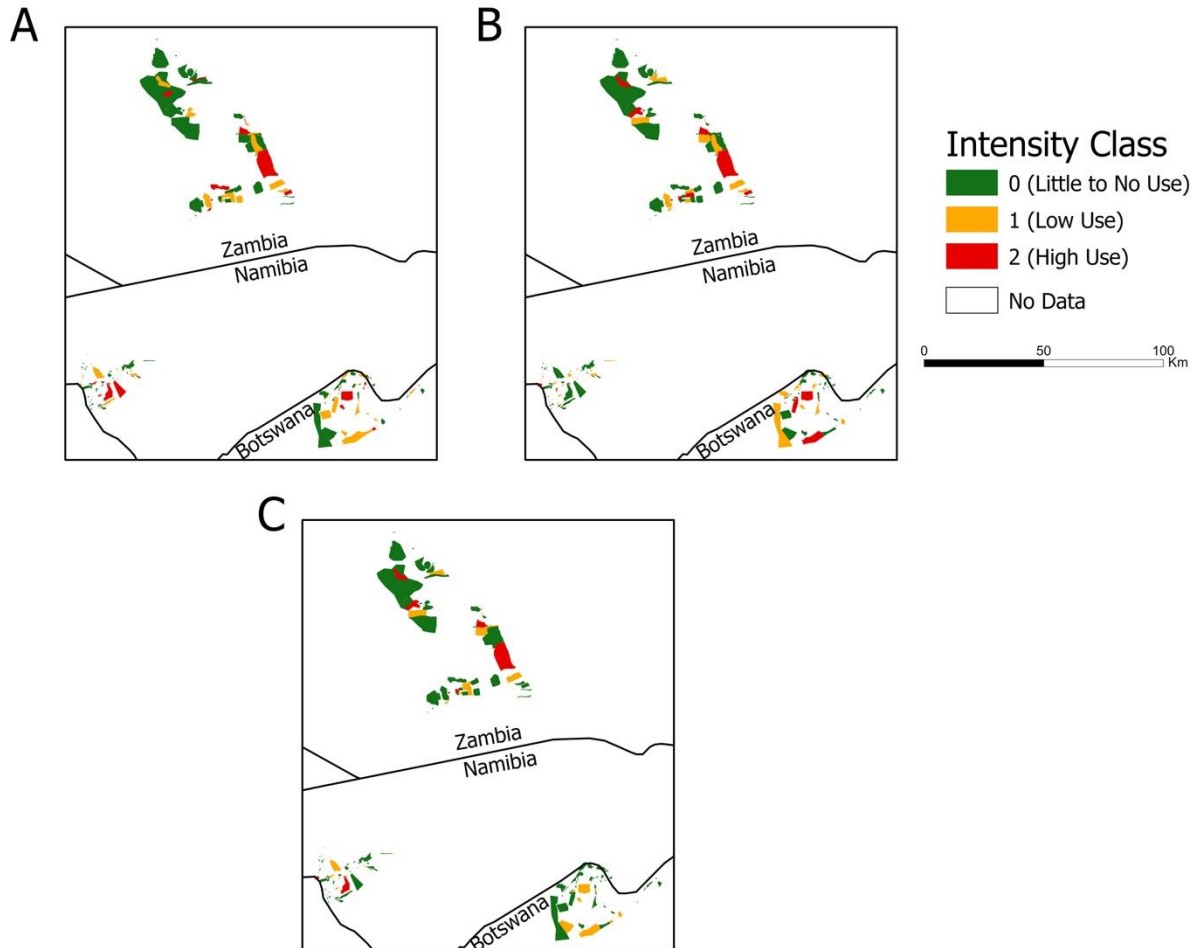

**Figure 3.** Resource use intensity labels assigned to pixels within each resource area for grazing (**A**), wood collection (**B**), and building pole collection (**C**). Country boundaries overlaid for reference.

To create Landsat-derived variables related to vegetation condition, we computed the Normalized Difference Vegetation Index (NDVI) and Soil-Adjusted Vegetation Index (SAVI) from the May/June image composites. Vegetation indices from these months specifically capture peak vegetation greenness at the onset of the dry season in this region (Figure S3) [30,31,92,93], providing model variables that represent the fully borne-out vegetated landscape that people engaged with in each image year. The NDVI is reliable in proxy estimates of aboveground biomass and photosynthetically active vegetation across a range of environmental monitoring applications [94–98]. The Soil-Adjusted Vegetation Index (SAVI) is used in conjunction with NDVI because of its utility in open canopy environments to mitigate background pixel brightness from dry soils [99]. In addition to

using the NDVI and SAVI for each year as model variables (M = 6), we created 6 more variables by performing an image difference calculation for each vegetation index between the three time-steps from 1994 to 2004, 2004 to 2018, and 1994 to 2018 using Equation (1):

$$Change_{VI} = VI_{T2} - VI_{T1} \tag{1}$$

where for a given vegetation index (*VI*), *T*1 is the earlier year and *T*2 is the later year (Figure S5).

Next, we applied the local Moran's I of spatial autocorrelation on the NDVI and SAVI of each year to provide measures of vegetation spatial patterns that may aid in boosting model performance [100,101]. Moran's I is a common spatial statistic for creating remote sensing texture features [102–104] and has been applied to NDVI in similar heterogeneous landscapes in southern Africa [105]. A bagged ensemble model such as random forest can benefit from additional information contained from texture features created with different focal window sizes but will be minimally affected by highly correlated variables [106,107]. Importantly, variable importance assessment will determine which window size, vegetation index, and year provided the most valuable information for accurate classification, which can be useful for feature selection and for making inferences about the most optimal scale by which to discern spatial landscape patterns [108].

We computed local Moran's I in R using the *MoranLocal* function in the *raster* package [109] for each vegetation index of each of our 3 years for 3 window sizes— 3 × 3, 7 × 7, and 11 × 11—similar to [110] (Figure S6). In total, we produced 30 Landsat covariates representing vegetation condition at the onset of the 1994, 2004, and 2018 dry seasons (Table S1).

### 4.2. Random Forest Modeling

We constructed pixel-based RF classifiers to predict resource use intensity of the five resource use types, using the pixels inside resource areas for training and validation. We assessed each set of RF models using users', producers', and overall accuracies. Additionally, we recorded variable importance for each model iteration. We utilized the *raster* and *randomForest* package in R [109,111] to construct random forests models. All training and validation protocols were consistent across model iterations and variants. Two important parameters are most commonly tuned by the practitioner—the number of input covariates to try at each tree node, and the number of trees to grow. In 2 comprehensive reviews on RF applications of remote sensing [65,70], the authors recommended that the number of trees per forest should generally be set to 500, which in most cases protects against overfit yet is computationally efficient. We followed accordingly and used the random forest algorithm's automated tuning function to choose the optimal number of variables per split for the lowest out-of-bag (OOB) error for every model.

Class imbalance was a consistent issue but was manifested differently between each resource type. To address this, we stratified the pixel observations by country and class value to use an area-proportional training sample allocation strategy supported by [69]. Because classes 1 and 2 were extremely rare in some resource types' response variables (Table A2), we merged them together into 1 sample group for the purpose of stratified sampling (Table A3). We then used the common RF training strategy of taking 70% of the pixels for training and 30% for validation [70,112].

We split modeling into 3 stages. In the first stage, we constructed separated RF classifiers using only Landsat covariates (M = 30) and only WorldPop covariates (M = 6) to assess prediction strength of each data source by their classification accuracies and to investigate the relative importance of each variable within the same source. Second, we combined all covariates together into RF classifiers to assess how all covariates behaved together and determine feature selection strategy. Finally, we performed feature selection to construct final random forests classifiers with an optimal set of covariates. Final prediction surfaces were then generated to produce summary statistics and the final resource use intensity maps.

Feature selection best practices for RF are under continuous investigation and are highly dependent on each dataset and application [113]. It is widely supported that the embedded variable importance ranking capability in RF yields similar or more useful feature selection results than external methods [66,68]. Still, the rationale behind how to use variable importance ranks differ widely and are decided upon on a case-by-case basis through data exploration. Because Landsat covariates were consistently less important than WorldPop covariates in the first 2 stages of modeling, we kept all 6 WorldPop covariates, then used the variable importance ranks from the Landsat RF models to select the top 5 most important (top 15%) Landsat features, similar to [114].

*4.3. Comparing Resource Use Intensity Patterns*

We calculated proportional area comprising each resource use intensity class for each CBO within the study extent in ArcGIS and described the areas of predicted resource use intensity in terms of the percentage of land predicted to each class. Last, we produced resource use prediction maps covering the study area, using the prediction surfaces generated from each final RF classifier. These maps help facilitate our discussion of model results, differences in resource use between the communities, and the challenges and considerations related to our methods that are applicable in the greater field of interdisciplinary remote sensing.

*4.4. Validating Model Outputs*

Because reference sample observations did not capture evidence of building pole collection, we exclude this model from the reference sample validation, instead focusing on grazing and firewood collection. We created 2 sub-groups of the field-collected reference samples: those whose field notes indicated evidence of grazing and those whose indicated evidence of firewood collection. We applied a 100-m buffer to each reference sample point, then calculated the majority pixel value from the respective prediction surface contained within the buffer. The majority pixel value along with photo validation allowed us to summarize, for grazing and firewood collection individually, how the model prediction surfaces spatially aligned with the reference sample observations.

## 5. Results

*5.1. Landsat RF Models*

Landsat RF classifiers performed poorly in overall accuracy across all resource use models, ranging from 61 to 71% (Table 3). Variable importance patterns differed for each resource type (Figures 4 and A1). In the grazing Landsat RF model, both 2004 vegetation indices and their $11 \times 11$ Moran's I texture derivatives exhibited high importance with a mean decrease Gini (MDG) value of 15 and higher (Figure 4). Across all resource use models, the $11 \times 11$ window size was more important than the other two window sizes derived from the same year's vegetation index in most instances, especially for those years in the top half of the variable importance rankings (Figures 4 and A1).

**Table 3.** Overall accuracies for each resource use type and model variant. Total variables available (M) were consistent for each model variant but total variables to try at each node-split (m) was tuned for lowest out-of-bag (OOB) error.

|  | All Variables | WorldPop | Landsat | Final Model |
|---|---|---|---|---|
|  | (M = 36) | (M = 6) | (M = 30) | (M = 11) |
| **Grazing** |  |  |  |  |
| m | 36 | 4 | 7 | 6 |
| Overall Accuracy | 91% | 94% | 64% | 92% |
| **Wood collection** |  |  |  |  |
| m | 22 | 4 | 7 | 9 |
| Overall Accuracy | 88% | 94% | 62% | 92% |
| **Building pole collection** |  |  |  |  |
| m | 33 | 6 | 15 | 11 |
| Overall Accuracy | 92% | 95% | 71% | 94% |

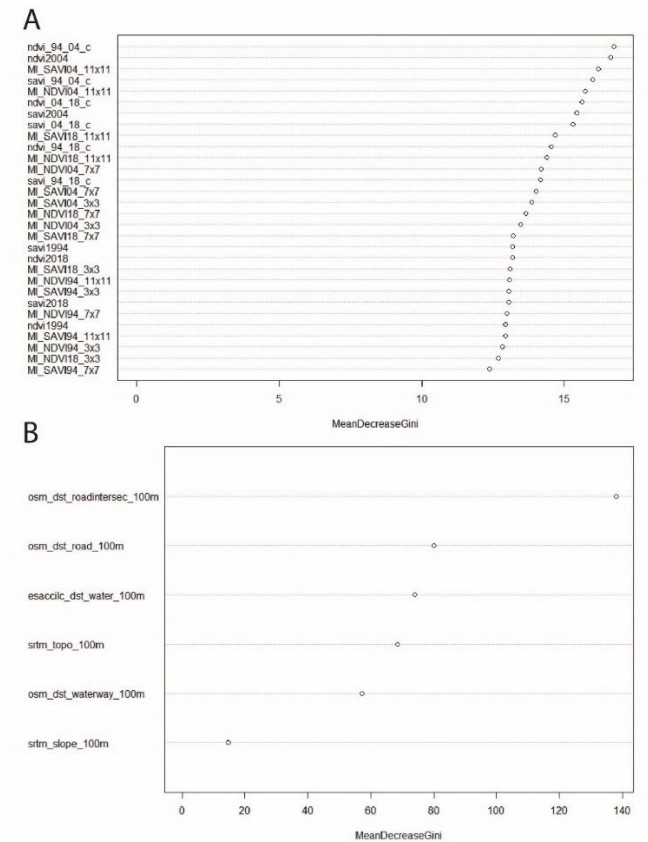

**Figure 4.** Variable importance plots for Landsat RF (**A**) and WorldPop RF (**B**) grazing intensity models.

The time-step change variables contributed strongly to all models, particularly for wood collection (Figures 4 and A1). The year 2004 was highly important across all models in various forms, especially as the NDVI and SAVI covariates in the building pole collection model (Figures 4 and A1), and Moran's I covariates in the grazing model. Class accuracies in the Landsat RF models were considerably different between the three intensity classes, especially for producers' accuracy. Across all resource use types, the lowest intensity

class (0) had high producers' accuracy, between 96 and 99%. This was compared to low producers' accuracies of class 1 and 2, ranging between 1 and 11%, and 10 and 16%, respectively (Table S2). Class 0 users' accuracies were much lower than its producers' accuracies, while classes 1 and 2 users' accuracies were much higher than respective producers' accuracies, with a difference ranging from 36 to 59% (Table S2).

### 5.2. WorldPop RF Models

Overall accuracies of WorldPop RF models were high, ranging from 93 to 95% (Table 3). The variable importance ranks of WorldPop covariates exhibited similar patterns between resource types, with distance to road intersections having highest importance and distance to roads ranking in the top three (Figures 4 and A2). Distance to water was ranked higher in the grazing models than in the other two models (Figures 4 and A2). Importantly, magnitude of variable importance was much higher in the WorldPop RF models compared to that of the Landsat RF models (Figures 4 and A2).

Producers' accuracies of class 0 were high like in the Landsat RF models. However, the user's accuracies of class 0 increased considerably, approximately 30–40%. Both accuracy metrics for class 1 and 2 improved in comparison with the Landsat RF models, with improvements from the Landsat RF metrics ranging from approximately 80 to 90% in producers' and 30 to 50% in users' accuracies (Table S3). All class-specific accuracies improved from the Landsat RF model, except for class 0 producers' accuracy, which remained above 95% (Table S3).

### 5.3. All-Covariates RF, Feature Selection, and Final RF Models

When all 36 covariates were included in the all-variables RF models, WorldPop covariates consistently received higher MDG values than Landsat covariates (Figure A3). In general, importance values for all variables differed little from their values in the Landsat RF and WorldPop RF models (Figure A3). Overall accuracies of the all-variables RF models were lower than WorldPop RF models, ranging from a 3 to 6% decrease (Table 3). The large discrepancy in MDG values between Landsat and WorldPop covariates led us to perform feature selection on Landsat covariates in the final RF classifiers.

The five highest ranked Landsat covariates from the Landsat RF models were selected to be included with the WorldPop covariates for the final RF models (Table S4). Among the most common Landsat covariates were ndvi2004, savi2004, and the NDVI and SAVI time-steps from 1994 to 2004 and 2004 to 2018. The 11 × 11-sized Moran's I of 2004 NDVI and SAVI were also in the top five for grazing and thatching grass collection.

Overall accuracies improved marginally from feature selection, with an improvement of 1–4% compared to the all-variables RF models (Table 3). However, overall accuracies of final RF models were still slightly lower than the WorldPop RF models, ranging from a 1 to 3% decrease in overall accuracy (Table 3). Class-specific accuracies in the final RF models were equal to, or slightly lower than, those of the WorldPop RF models (Table 4). Producer's accuracies declined by a range of 0–6%, while user's accuracy declined by 1–3% compared to the WorldPop RF models (Table 4). All final RF models produced overall accuracies above 90% and there was minimal difference in overall performance between the resource use types, within 2% (Table 3).

### 5.4. Predicted Resource Use Intensity

Grazing occurred at greater intensity over the largest amount of area in Mashi compared to the other CBOs, with almost equal amounts of land predicted to be grazed at low and high intensity (Figure 5). In the LWZ GMA, the least amount of intense grazing was predicted to be occurring, with 84.8% of the land having been assigned to the "little to no use" class (class 0). In CECT, low or high grazing intensity was predicted to be occurring over much of the areas covering the five surveyed villages.

**Table 4.** Confusion matrices for the final random forest (RF) models.

|  | Class | 0 | 1 | 2 | User's Accuracy |
|---|---|---|---|---|---|
|  | 0 | 17,276 | 823 | 585 | 0.92 |
|  | 1 | 259 | 5304 | 269 | 0.91 |
| Grazing | 2 | 124 | 150 | 4087 | 0.94 |
|  | Producer's Accuracy | 0.98 | 0.84 | 0.83 |  |
|  | Class | 0 | 1 | 2 | User's Accuracy |
|  | 0 | 16,548 | 778 | 571 | 0.92 |
|  | 1 | 503 | 5548 | 342 | 0.87 |
| Wood collection | 2 | 174 | 63 | 4350 | 0.95 |
|  | Producer's Accuracy | 0.96 | 0.87 | 0.83 |  |
|  | Class | 0 | 1 | 2 | User's Accuracy |
|  | 0 | 19,701 | 801 | 503 | 0.94 |
|  | 1 | 308 | 4297 | 52 | 0.92 |
| Building pole collection | 2 | 115 | 66 | 3034 | 0.94 |
|  | Producer's Accuracy | 0.98 | 0.83 | 0.85 |  |

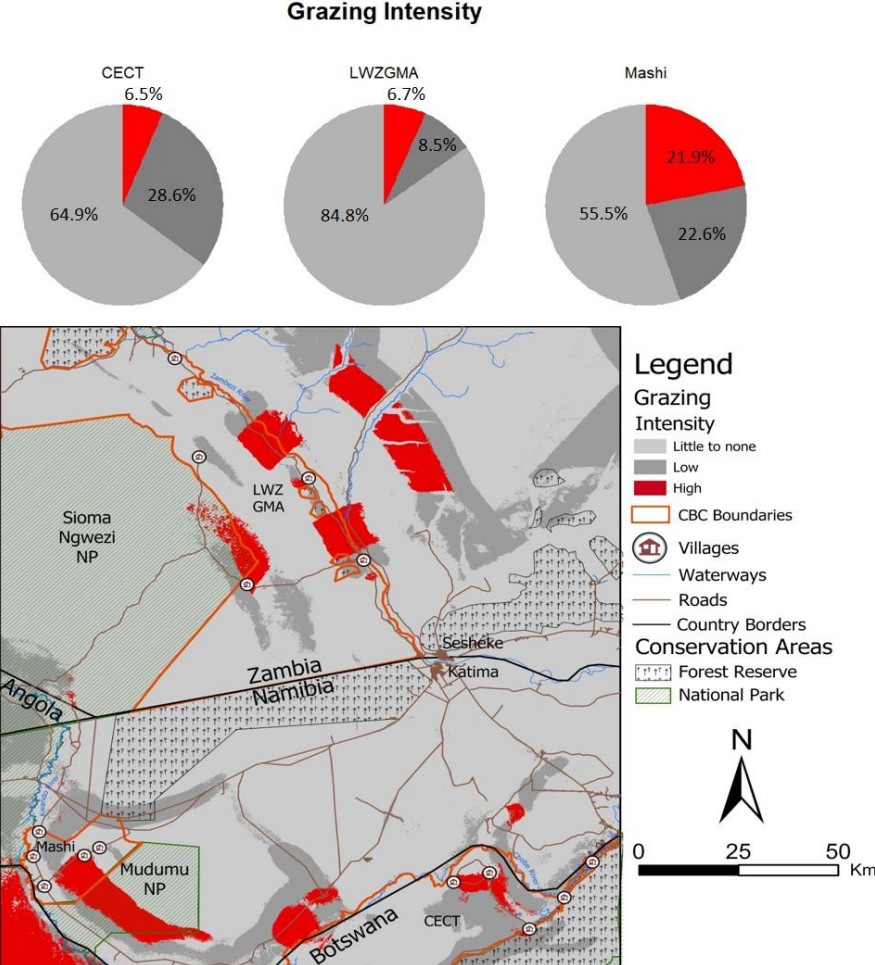

**Figure 5.** Grazing intensity prediction map and graphical summary.

Only 0.3% of the area in Mashi was predicted to have high wood collection intensity, with small amounts of high intensity wood collection in the other two CBOs; 9.9% and

5.8% predicted in CECT and the LWZ GMA, respectively. Over 60% of the land in CECT was predicted to have low wood collection intensity, with some of this land bordering the Chobe Forest Reserve. Almost all land in Mashi was predicted to have little to no wood collection (Figure 6).

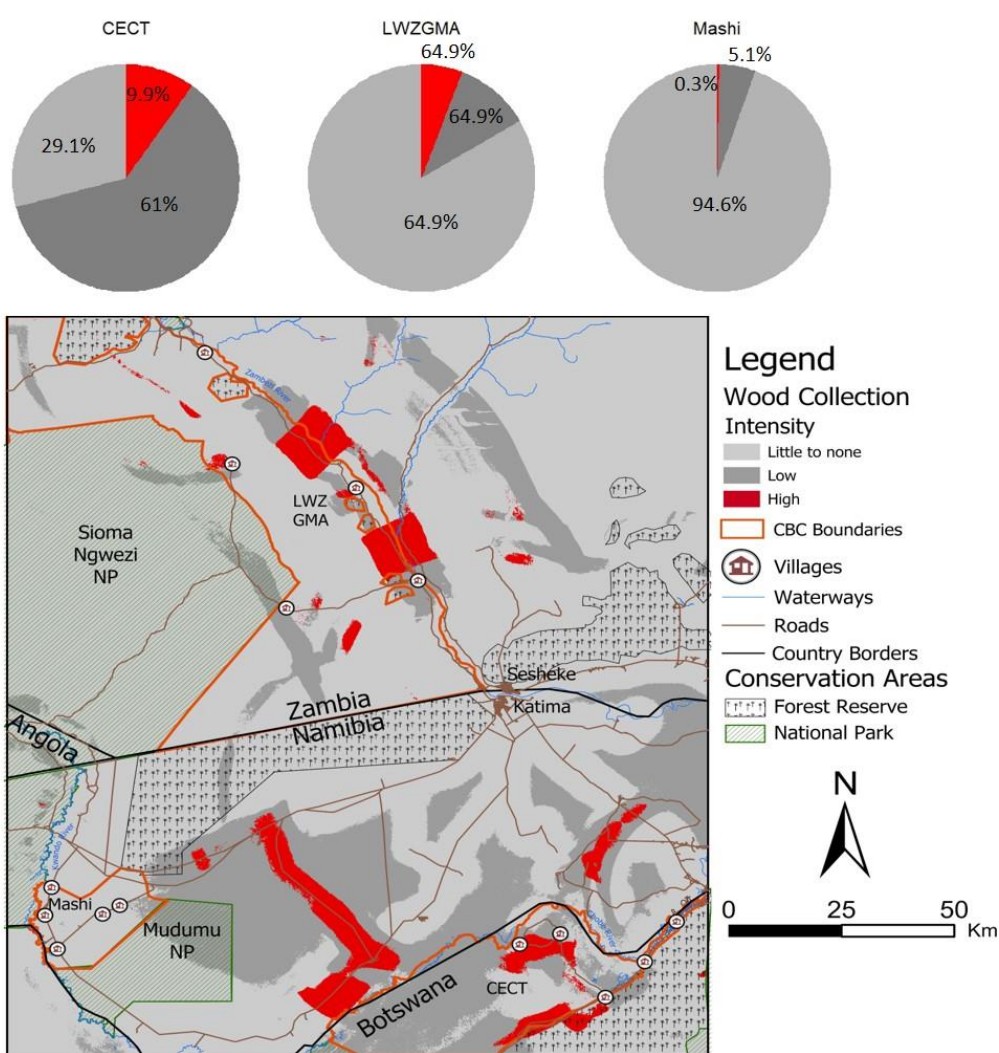

**Figure 6.** Wood collection prediction intensity map and graphical summary.

Overall, building pole collection was predicted to occur in relatively small proportions in all three CBOs, although the percentage of low and high use areas was largest in Mashi (Figure 7). In CECT, only 0.9% of land was predicted to have high building pole collection while 12.4% was to be under low use, mostly surrounding village centers, along roads, and abutting the Chobe Forest Reserve. In Mashi, most of the high building pole collection was in the southern portion of Mashi, toward Mudumu National Park (Figure 7). In LWZ GMA, high building pole collection was predicted to be occurring closer to the Kapau village center on the northeast boundary of Sioma Ngwezi National Park and in discrete patches of woodland far from roads and village centers.

**Figure 7.** Building pole collection prediction intensity map and graphical summary.

## 5.5. Validating Model Outputs

Of the total 198 reference samples, 135 samples indicated evidence of livestock grazing, while 75 samples indicated firewood collection evidence. Slightly less than half of the grazing-indicative reference samples were within low- and high-use prediction areas, while over 70% of samples indicating firewood collection were in "little to no use" predicted areas (Figure 8).

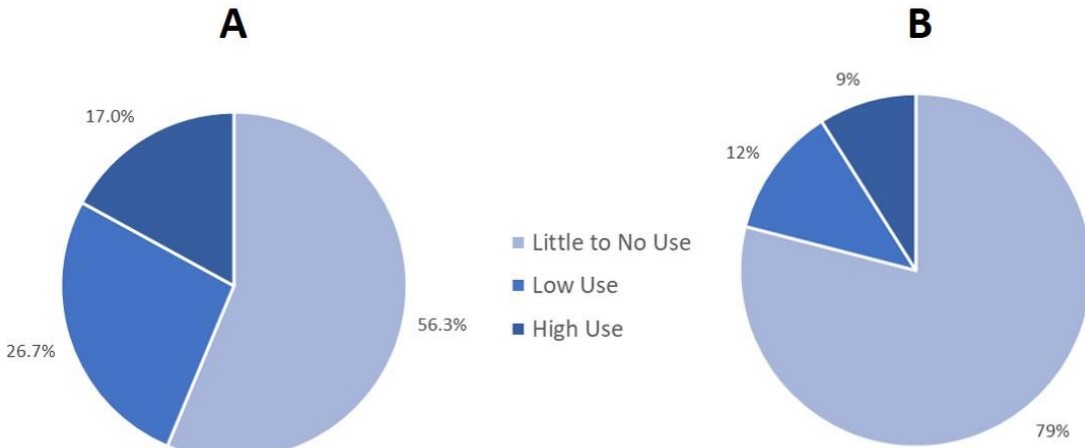

**Figure 8.** Percent of grazing-indicative (**A**) and firewood collection-indicative (**B**) reference samples falling within each predicted resource use intensity class.

## 6. Discussion

### 6.1. Population Proxy Predictors

From the high overall and class-specific accuracies of the WorldPop RF models (Table 3 and Table S3) and the high variable importance values of WorldPop variables (Figure A2), we can infer that the proximity and accessibility to resource areas from human settlements are extremely strong predictors for resource use intensity in our mapped resource areas, regardless of the resource use type [115]. Distance to road intersections and roads were two of the most important variables in all models (Figure 4, Figure A2, and Figure A3). Both provide important context regarding human accessibility to the landscape.

Although some roads may be narrow and unpaved, they nonetheless are cleared pathways into interior areas that would otherwise be difficult to travel through, an influential factor for natural resource extraction intensity in natural resource-dependent communities [115,116]. Proximity to denser populated areas is commonly associated with higher resource use intensity in natural resource-dependent regions [117,118]. Distance from road intersections provides important context, as human settlement in this region tends to be denser around road intersections in some instances, evidenced by several village center locations in each CBO and along roads (Figure 1 and Figure S7). The relationship between higher intensity of resource use and proximity to denser populated areas is illustrated in many of the final resource use intensity maps, where higher use intensity (classes 1 and 2) is predicted in clusters around road intersections and concomitant village centers (Figures 5–7).

### 6.2. Landsat Predictors

Overall Landsat RF accuracies were low in comparison to the other RF models (Table 3) and lower than the commonly cited 85% overall and/or producer's accuracy deemed acceptable [119,120]. Class 0 accuracies were highest and contributed most to overall accuracy (Table S2). However, the contrast in class 1 and 2 accuracies compared to class 0 illustrates that the Landsat RF models were largely unable to separate class signatures related to resource use intensity, rather they often predicted the majority class—"little to no use"—to optimize generalized accuracy, a common issue reported with machine learning and imbalanced datasets (Table A3 and Table S2) [65,70,121]. Although it is worth discussing why certain Landsat variables were more informative over others in each model, we are careful to draw direct conclusions about their utility in predicting resource use intensity given the stark performance differences between Landsat RF models and the other two RF model sets using the geospatial proxy variables.

The year 2004, represented in the NDVI, SAVI, time-step vegetation change, and Moran's I variables, was assessed by the model as more important than other years of Landsat data (Table S4), which is interesting given it is the middle year of the time series and also represents a point in time when all three CBOs had been formally established. During and after establishment of each CBO, land was allocated for tourism and for other conservation-related uses to facilitate freer wildlife movement [79,81]. Given that each CBO area has experienced population growth (Figure S7) and a reduction in usable land since 1994, the 2004 vegetation covariates may be discerning greater magnitude changes in areas of each CBO due to natural resource use intensification.

It is equally plausible that the discrepancies in rainfall timing and amount between years in the Landsat time series may be falsely exaggerating the year 2004's importance. Of the three years, rainfall was highest in March/April of 2004 (Figure S3). Given the well-documented 1.5–2-month lag between rainfall time and green up, especially in the floodplain areas [30,31,92,93], the 2004 covariates are possibly being ranked more important because of a greater contrast between productive vegetation and senesced or non-vegetated areas compared to the Landsat data from the other two years.

## 6.3. Feature Selection

Although the overall accuracies from the WorldPop RF models are slightly higher than the final RF models (Table 3), users' and producers' accuracies differed as little as 0–6% (Table 4 and Table S3). While the Landsat data did not contribute significantly to the mapping results, the variable importance values for Landsat variables in the all-variables RF models were not zero (Figure A3). Given that each forest grew 500 trees with which to produce the classification decision, some trees within the forest contained mostly (or all) Landsat variables and resulted in accurate classification. Otherwise, the variable importance values for these variables would be zero. Moreover, class accuracies of the Landsat RF models across all three intensity classes were above 50% (Table S2), indicating that there likely are pixels that were accurately classified partially or fully due to the information contributed by Landsat variables, which provide proxies of vegetation heterogeneity on the landscape.

The improvement of the final RF accuracies over the all-variables RF model, although minimal, justifies feature selection and inclusion of Landsat variables for the purposes simplifying a multi-source dataset model (Table 3) [122]. However, it also indicates that at a certain threshold of increasing dimensionality, including more Landsat covariates only adds more complexity with no aid to performance, often called the curse of dimensionality or the Hughes effect [123]. Because the difference in variable importance values between Landsat covariates beyond the several most important variables became vanishingly small (Figure 4, Figure A1, and Figure A3) and we understood that Landsat covariates were not contributing greatly to class accuracy, we selected the top five most important Landsat covariates for the final RF models. This demonstrates a common challenge encountered by practitioners, where specific feature selection choices are not clear-cut, nor is the result of feature selection consistent. Thus, this study is one among many that demonstrate the case-by-case nature of feature selection.

## 6.4. Final Model Performance and Resource Use Patterns

The 2% margin of overall accuracy differences between all resource use types (Table 3) and similarly small differences in class accuracies (Table 4) indicate that the final RF models were able to predict resource use intensity at similar rates of success, regardless of the resource use type. In general, spatial patterns of resource use intensity predictions were similar in more ways than they were different across resource types, in that higher intensity of resource use was concentrated near village centers and on either side of major roads in each CBO. This suggests that people's accessibility and proximity to core natural resource areas—as mediated by distance from homes, distance from roads, and topography—greatly

influence where natural resources are used and extracted. However, there were some differences in the amount of land area predicted to each intensity class between CBOs.

Mashi and CECT each had much larger predicted areas of low and high intensity grazing compared to LWZ GMA (Figure 5). Although this could be due to the dramatic size differences of LWZ GMA to the other two CBOs, it could be that greater proportions of land closest to surveyed villages are more ideal for grazing, especially in CECT (see Section 2). The most interesting pattern in the wood collection model was that Mashi had an overwhelming amount of little to no wood collection intensity predicted throughout (Figure 6). Because there were few resource areas reported to have high wood collection in Mashi compared to the other two CBOs (Table A2), it may be that people in Mashi more often collect wood near their homes or in the adjacent forest reserve instead of the mapped resource areas.

In addition to the common areas along roads and at village centers, high building pole collection intensity was predicted in discrete patches inside known forest reserves (Figure 7) and in other patches much further into the bush. Given that the building pole collection model improved using all five of the selected Landsat covariates, indicated by its *m* value in Table 3, vegetation condition may be more influential to spatial collection patterns than for other types of resources. This may be due to the more specific woody plant species sought after for building poles [124] and the unique growth or harvest pattern these areas would exhibit in satellite-derived vegetation indices over time [125].

### 6.5. Validating Model Outputs with Reference Samples

Results from the reference sample validation illustrate the complexity of modeling such nuanced, community-scale resource use patterns. A large percentage of firewood collection and grazing reference samples were within areas predicted to be under little to no use. Those reference samples that fell within areas predicted to be under low and high use exhibited land cover characteristics that are ideal for the given resource use, such as homogenous grasslands for grazing, or dense scrublands and woodlands for firewood collection. We were not surprised that reference samples and model predictions were less consistent for firewood than grazing. Firewood is a critical resource to rural people who use it daily and in great volumes. This reliance on firewood necessitates households to constantly gather firewood, which could occur in multiple resource areas or continuously throughout the day. Therefore, detecting a unique signal of intense firewood collection will be difficult, regardless if one is looking for an impact signal or a signal representing ideal areas for the activity [126,127].

### 6.6. Participatory Mapping and Remote Sensing

Meaningful community participation in land use and natural resource planning supports culturally relevant and equitable strategies that facilitate sustainable resource use [128,129]. Further, including local communities in such planning enables more robust climate adaptation and resilience through improved governance, trust building, legitimization of local knowledge, and social learning [130,131]. Participatory mapping, therefore, allows for the production of locally meaningful information that can be further incorporated with other spatial data for analysis-driven insights.

In our use-case, participatory mapping allowed us to probe the spatial relationships between resource collection behavior and spatial patterns of the human and physical landscape. Since all existing resource areas and resource collection activities could not possibly be mapped in the field for each community, we utilize the mapped resource areas as training data, similar to how other supervised land use/land cover classifications are performed. Given this framework, and with further efforts on remote sensing analysis, it should be possible to effectively map natural resource use activity both outside of mapped resource areas in each community and in other similarly communally managed areas nearby.

As the resolution and robustness of remote sensing analyses become more refined, the methods for collecting training and validation data must also be refined. Particularly in land systems science and socio-ecological research, the push continues to be made for fully integrated "people and pixels" approaches [34,37,49,53], but challenges remain in matching spatial and temporal resolutions of remotely sensed and social data [35]. In this study, we demonstrated an additional use case in which participatory mapping brought added value to the social data integration problem. Unlike other types of commonly available land use data, the spatial and tabular aspects of participatory mapping data are co-produced by researchers and community members. This allows researchers to ask questions whose potential answers are otherwise not verifiable, and in attempting to answer them, provides opportunities to develop new applications for remotely sensed data and to refine integrative analysis methods. Mapping NTFP collection and communal grazing intensity are now underway and will continue to rely on participatory mapping to benchmark the progress.

### 6.7. Challenges, Limitations, and Future Directions

To ensure that re-scaling the Landsat data to 100-m resolution did not distort or otherwise cause the model to lose useful information, we modeled grazing and wood collection at Landsat's native 30-m resolution, downscaling the WorldPop covariates from their native 100-m resolution to match. This produced little to no quantitative differences in accuracies, between 0 and 2%, and similarly negligible qualitative differences in the output prediction maps (Figure A4).

Of all freely available and low-cost imagery platforms, Landsat remains the priority because it allows landscape analysis over time periods stretching as far back as the 1980s, providing an opportunity to observe these community landscapes before they were incorporated into CBOs. Therefore, we believe the next step in this research pursuit should utilize greater than 12 Landsat images per year—or monthly composites—for every year in the current study's time range. From there, several directions can be taken. First, the Landsat spectral bands and additional texture derivatives from those bands and the vegetation indices should be incorporated into a similarly constructed model to determine if temporal data densification adds greater value to Landsat in the current standing methodology. Beyond this first experiment, another could produce multi-year land cover products and subsequent land cover change metrics such as in [132], then apply them to predict natural resource use intensity. A woody biomass change estimation could also be explored [133,134], necessitating field measurements.

Although they lack longer historical analysis capabilities compared to Landsat, higher-resolution imagery platforms that are freely available could produce greater success in mapping these natural resource use types and should be explored in the same progressive manner as with Landsat data discussed above. Recent studies have utilized a combination of Sentinel (10 m/pixel), PlanetScope (3 m/pixel), and other high-resolution imagery to answer questions related to land use and vegetation patterns. For example, the authors of [135] found a correlation of increased anthropogenic pressure to observed greening of woody vegetation and browning of non-woody vegetation in unprotected areas of the Greater Massai Mara, leveraging WorldView3 and Sentinel2 multispectral data. Additionally, the authors of [136] mapped land use–land cover types of smallholder agroecological zones in Kenya by incorporating spectral and texture metrics from PlanetScope, Sentinel-2, and Landsat image platforms into RF models. These studies support the value in exploring Sentinel-2 and PlanetScope platforms specifically, which are both freely available to researchers.

Grazing in this region likely fluctuates in intensity and location according to seasonal availability of pasture [137]. However, we did not account for seasonal differences of vegetation within years. Instead, we provided the model with one consistent timestamp per year at the onset of the dry season, which represents the time of year in which the most

grazing resources should be available. To solely focus on mapping grazing intensity, more intra-annual variation of vegetation should be incorporated into the model variables.

It was difficult to identify the best data standardization method for the model response variables, since the Household Use values derived from household surveys were of a non-normal distribution. Though we proceeded with the three-class ordinal response scheme for modeling, it is worth investigating other avenues such as regression. Class imbalance presented a challenge to the RF training process. Rarity of class 1 and 2 varied for each land use (Table A3), but in general one or both of these classes were too rare to allow for other class imbalance solutions to be feasible, such as over-sampling minority classes [121,138]. Proportionally sampling by class ensured that the training and validation sets were large enough, but we also stratified sampling by country to prevent unintentional exclusion of the rare class observations in Mashi and CECT (Tables A2 and A3).

The class rarity issue was illustrated in the wood collection predictions within Mashi (Figure 6). It is unlikely that little to no wood collection is occurring in the majority of Mashi, since firewood is a ubiquitous household necessity [126,127]. The household survey statistics demonstrated extremely low proportions of the resource areas reporting to be used at low or high intensity (Table A2). The RF model prediction underestimated these proportions more severely than in most other models because random forest often improves generalized accuracy by predicting the majority class [68,108]. The underlying cause behind such small proportions of class 1 and 2 wood collection being reported in Mashi was that two small resource areas had much higher household use than the rest, thus skewing the distribution of the Household Use value mean for the wood collection model (Figure 3; Table A2). This was likely related to Mashi community members relying on nearby bush and forest reserve land for firewood collection, with the exception of those select resource areas.

We did not map all resource areas that were listed on the surveys due to time and geographic constraints. Therefore, we acknowledged that our models were trained only with the variation of the landscape captured by the sampled areas. Additional natural resource areas and resource use patterns exist in each CBO (and are estimated by models outside of our mapped areas) but were not reflected in the training data and the subsequent model predictions. This offers an opportunity for additional future field work to validate these predicted use areas. The overall spatial distribution of mapped resource areas represents a clustering around village centers and roads; thus, the importance of proximity and accessibility to resources for communal land use is unsurprising and intuitively makes sense.

The nature of using participatory mapped data as training and validation data means that model accuracies reflect only how well the models predicted resource use intensity within the resource areas. Therefore, we do not condone the use of these maps for land management outside of each CBO or in national parks and forest reserves. Last and most importantly, we stress that any and all results from this research should be used cooperatively by the local community members in each CBO and their local government agencies. They should not be used as evidence-based support for top-down land management decisions made without consent of the local communities. This analysis does not fully reflect the complex land use history within each CBO area, nor does it take into account implications of particular management policies as they relate to resource use activity on the landscape.

## 7. Conclusions

This study presented a unique integration of remote sensing methods with participatory mapping to map spatial patterns of natural resource use intensity in a communally managed African landscape. We compared the utility of Landsat data and WorldPop geospatial covariates in their ability to classify various non-timber forest product (NTFP) collection and grazing activities. This study progresses the remote sensing methods commonly used in such integrated remote sensing studies by outlining a framework of in-

tegrated, multi-source geospatial modeling, as well as by assessing the advantages and challenges of using participatory mapped training data in a supervised machine learning-based classifier. We make several important conclusions from this work:

(1) The covariates we derived from Landsat data were limited on their own in providing an accurate model of resource use intensity in the current modeling methodology, but Landsat may remain the best platform for this task because of its long operational timespan. To further inquire into the utility of Landsat in this classification application, we first suggest extracting variables from a denser time series (>12 images for every year), using spectral bands and more textural metrics in a similarly constructed classification model. If this is not fruitful, we suggest exploring other Landsat-based approaches such as multi-year land cover change metrics or woody biomass proxies to discern relevant vegetation trends for each natural resource activity. Additionally, the utility of higher-resolution, freely available imagery platforms, such as Sentinel 2 and PlanetScope, ought to be explored in similar methodological frameworks.

(2) Covariates reflecting proxies of population density and mobility proved to be much more powerful than expected and were responsible for the majority of model performance. This supports our fundamental argument that in communally managed landscapes such as these, patterns of accessibility and proximity to important natural resource areas are strong predictors of where people will use natural resources and at what level of intensity. Therefore, ancillary geospatial covariates such as the ones used in this study should continue to be given high consideration when attempting to map resource use intensity in rural heterogenous landscapes.

(3) The spatial patterns of resource use intensity surprisingly differed little between resource use types, which we relate to the same common rules of the human landscape being applied to resource use regardless of the resource type. Pattern differences of resource use intensity between CBOs were therefore largely a reflection of the patterns in which humans are settled within each community and the ways that they can most efficiently travel.

(4) Challenges were numerous in deriving training data from the participatory mapped resource area dataset. These challenges and our proposed solutions contribute insights to the feasibility of integrating supervised machine learning-based remote sensing analysis with participatory mapping and other survey-derived datasets going forward.

(5) The results from this study suggest that mapping NTFP collection and grazing intensity at the community scale pose a relatively new challenge for remote sensing practitioners, but that participatory mapping can provide useful training and validation data for the analysis process.

The integration of participatory mapping expands the breadth of unique human–environmental questions that researchers can apply remote sensing, which will inevitably lead to innovations in remote sensing analysis techniques. Key insights from this study can be used to further progress the integration of remote sensing for socio-ecological systems, especially those that attempt to use participatory mapping and remote sensing technology and methodologies. Further work should continue to investigate the capabilities of satellite remote sensing in finer spatial and temporal resolutions and will especially take more ancillary geographic variables into consideration. The maps are not intended for official interpretation outside each CBO, and they should not be used in any way that excludes or undermines community members within these CBOs.

**Supplementary Materials:** The following are available online at https://www.mdpi.com/2072-429 2/13/4/631/s1: Figure S1: ESA Copernicus 2018 Land Cover composition of each CBO. Figure S2: Landsat footprints. Figure S3: CHIRPS rainfall timeseries. Figure S4: WorldPop population proxy covariates displayed geographically. Figure S5: Vegetation index change variables displayed geographically. Figure S6: Moran's I texture derivative example displayed geographically in a subset area. Figure S7: Population density change estimates for 2000–2018. Table S1: Landsat-derived model

variable names. Table S2: Confusion matrices for Landsat RF models. Table S3: Confusion matrices for WorldPop RF models. Table S4: Landsat variables selected for final RF models.

**Author Contributions:** Conceptualization, K.D.W., N.G.P., F.R.S., and A.E.G.; data curation, K.D.W. and K.M.B.; formal analysis, K.D.W.; funding acquisition, N.G.P. and A.E.G.; investigation, K.D.W., A.E.G., N.E.K., M.D.D., J.S., L.C., J.H., and H.M.L.; methodology, K.D.W., N.G.P., and F.R.S.; project administration, A.E.G., J.S., L.C., J.H., and H.M.L.; writing—original draft, K.D.W.; writing—review and editing, K.D.W., N.G.P., F.R.S., A.E.G., N.E.K., M.D.D., J.S., L.C., J.H., and K.M.B. All authors have read and agreed to the published version of the manuscript.

**Funding:** This research was funded by National Science Foundation, grant number 1560700.

**Institutional Review Board Statement:** Ethical approval for the collection of data used in this study was approved by the University of Colorado Institutional Review Board (#16-0126). Permission to conduct research was granted by the traditional authorities and local leadership in all study communities.

**Informed Consent Statement:** Informed consent was obtained from all subjects involved in the study.

**Data Availability Statement:** The data presented in this study are openly available at https://earthexplorer.usgs.gov/ and https://www.worldpop.org/project/categories?id=14.

**Conflicts of Interest:** The authors declare no conflict of interest.

## Appendix A

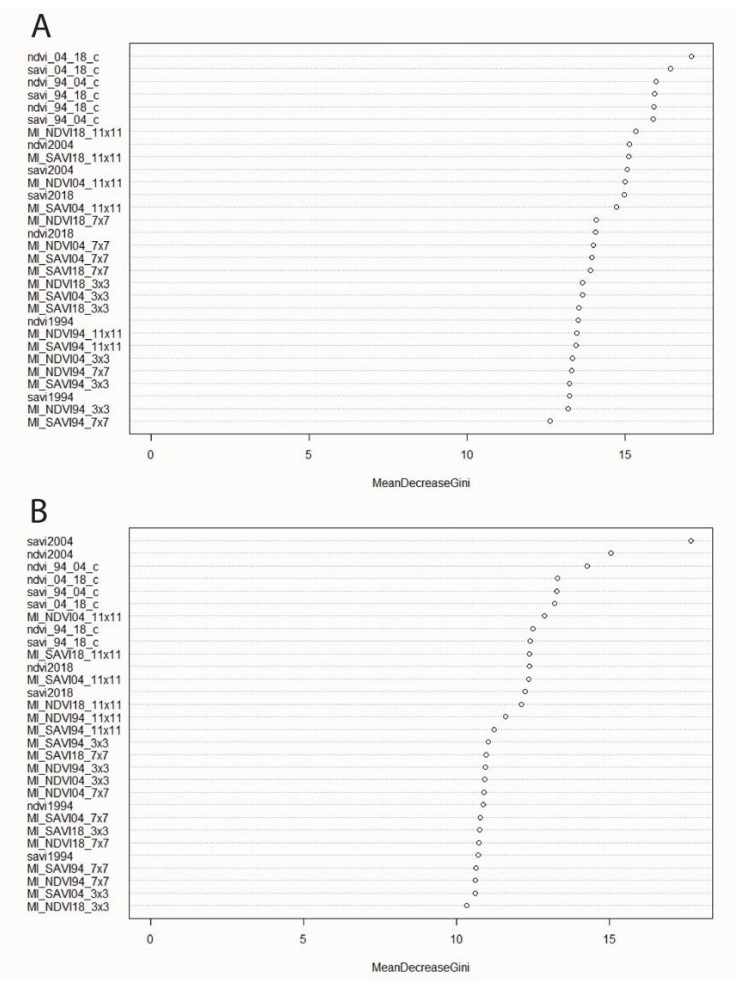

**Figure A1.** Variable importance plots for Landsat RF wood collection (**A**) and building pole collection (**B**) models.

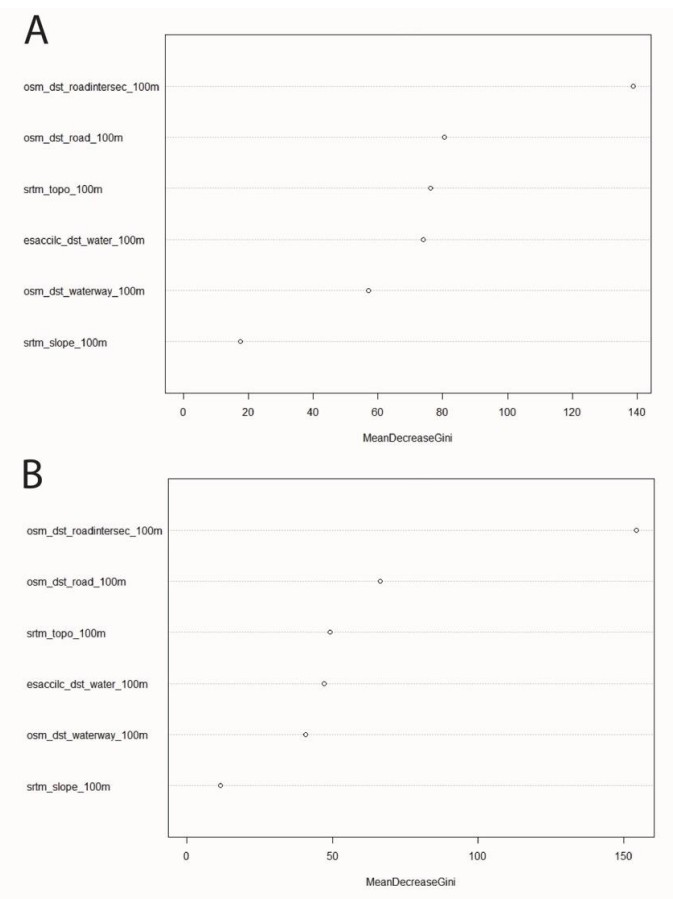

**Figure A2.** Variable importance plots for WorldPop RF wood collection (**A**) and building pole collection (**B**) models.

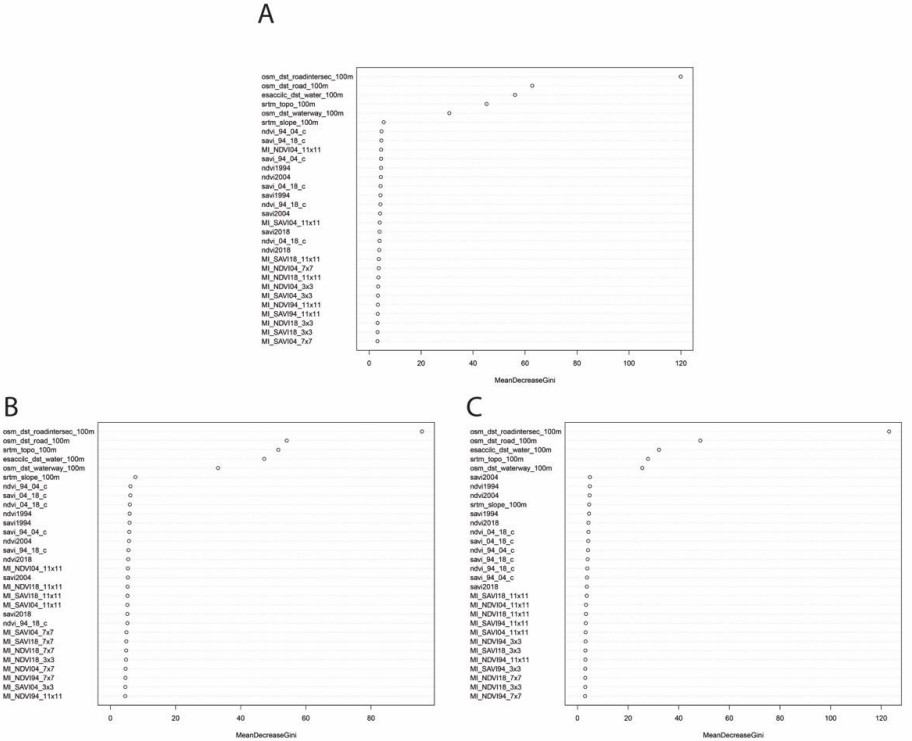

**Figure A3.** Variable importance plots for All Covariates RF models: grazing (**A**), wood collection (**B**), and building pole collection (**C**).

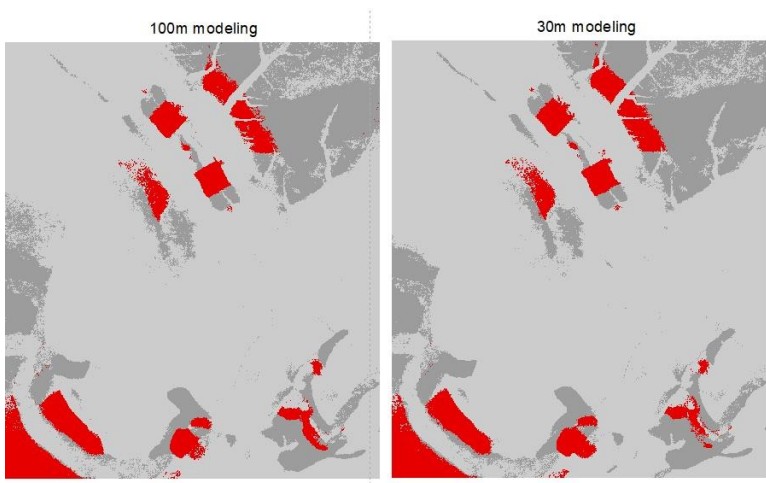

**Figure A4.** Grazing intensity prediction outputs side-by-side at 100 meter and 30 meter resolution.

**Table A1.** Percent land cover within each CBO area, according to ESA Copernicus 2018 Land Cover Product.

| LAND COVER | CECT | MASHI | LWZ GMA |
|---|---|---|---|
| SHRUBS | 60.5% | 27.2% | 13.1% |
| HERBACEOUS VEGETATION | 7.0% | 1.0% | 2.2% |
| CROPLAND | 0.4% | 3.2% | 1.7% |
| URBAN/BUILT AREA | 0.2% | 0.0% | 0.1% |
| WATER BODIES | 0.1% | 0.0% | 0.6% |
| HERBACEOUS WETLAND | 9.3% | 2.7% | 0.9% |
| CLOSED FOREST | 1.3% | 0.6% | 5.8% |
| OPEN FOREST | 21.2% | 65.2% | 75.7% |

**Table A2.** Percent area of resource area training dataset by each CBO and resource use intensity class.

| | Class | CECT | Mashi | LWZ GMA |
|---|---|---|---|---|
| Grazing | 0 | 10471 (50.2%) | 2573 (31.3%) | 48,740 (69.8%) |
| | 1 | 8000 (38.3%) | 2262 (27.6%) | 10,768 (15.4%) |
| | 2 | 2398 (11.5%) | 3375 (41.1%) | 10,315 (14.8%) |
| Building Pole Collection | 0 | 13862 (66.4%) | 5057 (61.6%) | 48,872 (72.6%) |
| | 1 | 6984 (33.5%) | 1618 (19.7%) | 8228 (12.2%) |
| | 2 | 20 (0.1%) | 1532 (18.7%) | 10,226 (15.2%) |
| Wood Collection | 0 | 6942 (33.2%) | 6807 (82.9%) | 46,074 (66.6%) |
| | 1 | 8406 (40.3%) | 1086 (13.2%) | 11,731 (17.0%) |
| | 2 | 5536 (26.5%) | 317 (3.9%) | 11,327 (16.4%) |

**Table A3.** Training pixel counts within each resource use intensity class and each country. Bolded value pairs of class 1 and 2 in each country group were merged before stratified sampling.

| Country | Class | Pixel Count |
| --- | --- | --- |
| 1 | 0 | 10,540 |
| 1 | 1 | **7896** |
| 1 | 2 | **2475** |
| 2 | 0 | 2978 |
| 2 | 1 | **2766** |
| 2 | 2 | **3500** |
| 3 | 0 | 45,349 |
| 3 | 1 | **10,284** |
| 3 | 2 | **10,470** |

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
