# Peer review of "Modeling Community-Scale Natural Resource Use in a Transboundary Southern African Landscape: Integrating Remote Sensing and Participatory Mapping"

_remotesensing, doi:10.3390/rs13040631_

Round 1
Reviewer 1 Report
Very happy with the changes that have been made. Happy to see it published.
Author Response
Thank you for your time and meaningful feedback throughout the review process.
Reviewer 2 Report
Thank you for your revisions.
The only comment I have left is about the abstract, which seems a bit overly optimistic considering the final results. Please highlight that:
- resource use intensity mapping remains a challenging task (and shortly frame why)
- the contribution of the Landsat data to the mapping results was very limited and further research is needed to augment the added value of this data source for resource intensity mapping
Author Response
Thank you for your time and insightful suggestions thus far. We have modified the abstract per your recommendations to adjust the tone about the modeling results and to explain why we think mapping resource use intensity is challenging in this study region. You will find these changes on lines 19-22 and 32-39 in the revised manuscript.
Reviewer 3 Report
Thanks for the point-to-point response to my comments. The authors have emphasized the contribution of the work in the resubmitted manuscript. I agree with the authors that the proposed model has a new insight. But I insist that it is also essential for a research paper to provide robust experiment results or convincing examples, at least within a small study area or specific data, to support the main idea. Otherwise, the contribution and potential of the proposed method would be untenable.
For this paper, like the concern of the other reviewer, the experimental results suggest that it is absolutely unnecessary to integrate remote sensing images with the model. So the question this paper tries to answer is still hanging in there: is the integration of remote sensing data and geospatial proxy data really effective? In my opinion, although the proposed method was claimed to be novel and with a huge potential contribution, the authors failed to provide an example to support the proposed method. In fact, the experiment gave a reverse result showing the integration of Landsat data will reduce the mapping performance.
The authors repeated many times in the response letter as well as the paper that they introduced a new concept. But the experiment was not improved at all in the revised manuscript. Again, the authors must present a sound experiment with positive results to demonstrate the effectiveness of the proposed method, rather than just publishing some new insights or concepts.
Author Response
We very much appreciate the time and consideration that the reviewer has given the paper, and acknowledge that there are only minor misconceptions on their part regarding our results (e.g. it would be a mischaracterization of the modeling methodology to conclude that the inclusion of remotely sensed variables hurts the modeling result). We agree with the reviewer that there should be substantiated findings to support conclusions of our work, and believe we have provided several, albeit not one that unequivocally determines Landsat’s effectiveness in this particular mapping task. We humbly point out that the assumption the reviewer is implying here, that publishable results should be both novel and confirmatory, is unsound. What we show, and what we argue is highly relevant to the remote sensing literature, is that our assumption that remotely sensed proxies of land cover would be informative in modeling resource uses is incorrect, given the methods and data employed.
As remote sensing and land systems scientists, we devised the current methods under these assumptions and found unexpected results regarding the relative contribution of data sources to the modeling. Had a paper like ours already been available in the literature, with the detailed caveats and limitations we’ve now included thanks to reviewer responses, we may have avoided such assumptions and moved on to more refined methods. Negative or less than confirmatory results are still results and we hope that future readers of this article will find value in the nuances found therein. Understanding the limitations of remote sensing data in different contexts and methods is, in our opinion, just as important as knowing only what works. We therefore would ask for consideration from the editors that given the additions within the discussion, which acknowledge the various reasons why remotely sensed data were not as informative as we anticipated, that the work is still of interest and important to publish.
Our approach is worthy of critique, and the reviewers’ suggestions and contributions to the discussion of such methods have been well received for future research. Along with the request from Reviewer 2, we have made some additional edits in the manuscript to tone down the optimism of the findings in the abstract to more accurately reflect points already made in the discussion. Importantly, we note the relevance of our study in a reputably difficult landscape to model when relying on spectral signatures of remote sensing data alone in the abstract.
Round 2
Reviewer 3 Report
This is the third version of the manuscript. But the experiment has not been improved at all. Although several reviewers have questioned the result and given potential solutions, I do not think the authors have tried to conduct any new experiment accordingly. Like I commented in the last round of review, the effectiveness of integrating remote sensing imagery with geospatial proxy data is still questionable. For a research article, the author should answer the question with sound and comprehensive experiment results. But I did not see any substantive improvement in the revised manuscript.
For example, in the first round of review, I doubted the usefulness of Landsat or other remote sensing imagery, such as Sentinel-2 or MODIS. The author responded in the paper that “the contribution of Landsat data as utilized in our modeling framework was negligible, and further research must be conducted to extract greater value from Landsat or other optical remote sensing platforms to map these land use patterns.” This should be a simple comparison and easy to implement, just replacing Landsat with other data. The conclusion, positive or negative, would be significantly strengthened with these results. But I do not understand why the authors refuse to conduct any new experiment. Compared with the original submission, the only major revision is the additional edits in the discussion and future works.
I acknowledge that the paper has a new insight. However, the quality of the paper, either methodology or experiment, has not been improved. I think the analysis of remote sensing data is still limited. More importantly, the conclusion is still untenable without substantiated experiment results.
Author Response
We thank Reviewer 3 for their feedback thus far and respect their opinion regarding our study. We disagree that the conclusion is untenable, as there were additional valuable conclusions drawn from this work aside from the question of Landsat's effectiveness at the mapping task. However, we look forward to improving the remote sensing analysis methods in further research, taking several of the reviewers suggestions under consideration. The editors have decided on acceptance after making several revisions. We have copied our line-by-line response to the editors here for the reviewers knowledge.
In the Discussion section, lines 574-597, we have expounded on the necessary next steps for this research, providing specific prescriptions for the Landsat platform and relevant papers that utilize Sentinel, PlanetScope, and other high-resolution platforms for similar mapping tasks. In the conclusion, lines 654-663, we have amended the first numbered conclusion point to echo these new discussion points.
Lines 251-256 in the Methods section have been modified to explain the seasonal pattern of vegetation growth in the study region and how we used this knowledge to intentionally select May/June imagery for the calculated vegetation indices.
Lines 598-603 have been added in the discussion to acknowledge the fact that spatio-temporal variation of grazing intensity likely occurs with seasonal availability of grazing resources. We note this as a limitation of the current study’s methodology and suggest that intra-annual vegetation proxies be incorporated into future grazing intensity models.
In the Discussion section, lines 555-567, we highlight how participatory mapping can improve remote sensing methods as applied to land systems science questions.
In the Conclusion, lines 685-690, a fifth conclusion point has been made to end the paper’s narrative on the importance of participatory mapping in improving remote sensing methods.
Figure 2 has been moved to the Supplementals (now Figure S1) and Table 2 has been moved to the Appendix (now Table S4).
In addition to the two in-text items, we have moved a total of 3 tables and 4 figures from the Supplementals to the Appendix. We now feel that all current figures and tables presented in-text in the manuscript serve a purpose, but if the editors would like any additional figures or tables moved to the supplementals, we ask that they please reference them in the request.
We would like to thank all editors and reviewers for the attention they have given to the manuscript thus far.